# Recent changes to Arctic river discharge

Dongmei Feng [1✉], Colin J. Gleason[1], Peirong Lin [2], Xiao Yang [3], Ming Pan [2,4] & Yuta Ishitsuka[1]

Arctic rivers drain ~15% of the global land surface and significantly influence local communities and economies, freshwater and marine ecosystems, and global climate. However, trusted and public knowledge of pan-Arctic rivers is inadequate, especially for small rivers and across Eurasia, inhibiting understanding of the Arctic response to climate change. Here, we calculate daily streamflow in 486,493 pan-Arctic river reaches from 1984-2018 by assimilating 9.18 million river discharge estimates made from 155,710 satellite images into hydrologic model simulations. We reveal larger and more heterogenous total water export (3-17% greater) and water export acceleration (factor of 1.2-3.3 larger) than previously reported, with substantial differences across basins, ecoregions, stream orders, human regulation, and permafrost regimes. We also find significant changes in the spring freshet and summer stream intermittency. Ultimately, our results represent an updated, publicly available, and more accurate daily understanding of Arctic rivers uniquely enabled by recent advances in hydrologic modeling and remote sensing.

[1] Department of Civil and Environmental Engineering, University of Massachusetts, Amherst, MA, USA. [2] Department of Civil and Environmental Engineering, Princeton University, Princeton, NJ, USA. [3] Department of Earth, Marine and Environmental Sciences, University of North Carolina at Chapel Hill, Chapel Hill, NC, USA. [4] Center for Western Weather and Water Extremes, Scripps Institution of Oceanography, University of California San Diego, La Jolla, CA, USA. ✉email: dongmeifeng@umass.edu

Arctic rivers supply water and energy for over 50 million people[1] and convey freshwater, heat, and terrigenous material (e.g., sediment, nutrients, and carbon) to regulate the biological productivity of both inland and coastal ecosystems[2,3]. Arctic river freshwater impacts the thermohaline circulation in the North Atlantic, which further affects the global climate as a vital part of earth's general circulation[4]. Here, the term Arctic river refers to all rivers eventually draining into the Arctic Ocean, Bering Strait, and the Hudson, James, and Ungava Bays (HJUBs) and excluding the Greenland Ice Sheet, with a total drainage area of 22.1 million km². Previous studies have shown that Arctic hydrology is disproportionally affected by climate change due to the high sensitivity of its cryospheric components to climate warming and Arctic amplification[5,6]. Therefore, alterations in Arctic rivers can significantly impact societal and ecosystem functions, feedback with the global climate, and create increased uncertainty for future global climate conditions.

River discharge (a.k.a. flow rate or streamflow, the volumetric flux through rivers per unit time) integrates all hydrologic processes of upstream watersheds, defines a river's carrying capacity, and is perhaps the single most important measurement needed to understand a river[7]. However, our knowledge of Arctic river discharge remains limited due to a lack of trusted, comprehensive, and publicly available data. Stream gauging is a mature technology to monitor river discharge automatically, but the translation from automated stage measurements to river discharge estimates is susceptible to errors stemming from out of bank flows, shifting channel geometry, poor under-ice calibration, infrequent site access, and most importantly the often violent ice breakup period[7,8]. Furthermore, publicly available and up-to-date gauges in the pan-Arctic region have declined dramatically since the mid-1980s[9–11]. Domestic and international politics partially lead to this gauge decline, and we acknowledge that our use of satellite data is an active circumvention of those politics. For a period of rapid warming 1984–2018[12], there are approximately 1,300 Arctic gauges with publicly available daily data, and only 293 (23.2%) of these are more than 90% temporally complete. This gauge monitoring system is also spatially biased, with more than 89% of these gauges located in North America and all gauges subject to data access policies of the organization that maintains them. While previous studies[6,13–16] have investigated recent changes in Arctic hydrology under climate change, they have mostly relied on these biased gauge data. Therefore, a spatially and temporally comprehensive picture of recent changes in pan-Arctic river discharge is absent in our current knowledge.

It is now possible to provide a spatially and temporally comprehensive picture of river discharge by combining gauges with orbital satellites and hydrologic modeling. Satellites provide needed primary data to improve understanding of Arctic hydrology[7], especially considering recent breakthroughs in remotely sensed discharge inversion algorithms[17–19], development of global hydrologic datasets[20–22], and advances in data availability and computational resources for remotely sensed data[23,24]. Remotely sensed data, however, are sparse in space and time by definition given orbit geometry, and thus the best way to better understand Arctic rivers without new gauges is to combine remote sensing and hydrological modeling. Although recently modeled global river discharge products show promising skill in the Arctic[25,26], fusing models with remote sensing would allow the primary data of remote sensing to drive improvements in hydrologic models, which would then propagate the information gained from satellites in space and time to all rivers using classic hydraulic physics[27]. Further, improvements to hydrologic models gained from incorporating remote sensing can be used to reduce uncertainty in predictions of the Arctic hydrologic state[28].

In this study, we combine remote sensing and models to produce a discharge product that is spatially and temporally complete for the entire pan-Arctic: the Remotely-sensed Arctic Discharge Reanalysis (RADR). RADR is a record of daily discharge for 486,493 river reaches across the pan-Arctic region over the period 1984–2018. We generated RADR by assimilating approximately 9.18 million discharge observations derived from 227 million river width measurements from Landsat images (Supplementary Fig. 1) into an optimal blend of two recent global hydrologic model simulations[25,26]. See Methods on how we generate discharge from Landsat[19], the global hydrologic models we used to drive the assimilation[25,26], and our data assimilation scheme[27]. All of our methods are based on techniques and model results described in recent literature.

## Results

RADR shows that the Arctic is exporting more water to the Arctic Ocean than previously reported, and reveals significant changes to spring freshet timing and stream intermittency over the past 35 years (Fig. 1). Based on RADR, we calculate the average freshwater export to the Arctic Ocean as 5,169 km³/yr, ranging from 4,656–6,073 km³/yr, with North American rivers contributing 1,768 km³/yr (34.2%) and Eurasian rivers contributing 3,401 km³/yr (65.8%). Average runoff (discharge normalized by drainage area) to the Arctic Ocean was 234 mm/yr, which is 3–17% higher than previous estimates[29] (runoff allows direct comparison with previous less spatially complete studies via area normalization). Further, RADR shows a significant positive acceleration in discharge ($p < 0.05$) of 11.6 km³/yr/yr (0.22%/yr) across all rivers and 3.4 km³/yr/yr (0.19%/yr) for rivers in North America (Fig. 1b). This pan-Arctic acceleration (11.6 km³/yr/yr) is a factor of 1.2–3.3 greater than previous estimates (3.47–10 km³/yr/yr)[29–34] (see details in Supplementary Table 1). These trends hold at the pan-Arctic scale, but RADR does show many individual rivers, and some entire regions, opposing these increasing trends as revealed by RADR's spatial completeness and resolution, as discussed later.

RADR is spatially and temporally complete, and therefore differences in total water export and acceleration partially come from accounting for more rivers than previous literature via modern remote sensing. Supplementary Table 1 compares previous literature[6,16,30–32,35,36] with RADR explicitly, in which we calculate RADR results to match previous work in space and time where possible. We find that RADR shows increases in total water export and acceleration across many of these comparisons, showing that RADR's insights are not limited to its extended spatial coverage. However, RADR's extended spatial domain is perhaps its strongest feature. There are only 293 out of 486,493 RADR reaches that have a temporally complete daily gauge record for our study domain/time period, and 93% of these reaches are in North America. Furthermore, smaller rivers dominate the pan-Arctic hydrography (364,745, or 75% reaches in total are found in stream orders 1 and 2, as expected following the Horton's Laws[37]), but these smallest rivers are the least gauged (Supplementary Fig. 2) and frequently not included in previous literature[30–33,38]. While we cannot verify RADR flows at ungauged basins by definition, RADR is unique in providing spatially and temporally explicit flows at all Arctic rivers.

This completeness across the pan-Arctic allows unprecedented spatial analysis (Fig. 2). We found that 22.7% (15.8% with the field significance test[39]) of our 486,493 pan-Arctic reaches show a significant trend in discharge. Previous studies[30–33] have shown increasing discharge trends, but none allows the level of spatial detail RADR provides (Fig. 2). Most reaches showing significant trends (82.5%) are small streams with a drainage area less than 1000 km², which are unlikely to be included in previous studies using only gauge datasets to understand Arctic hydrology[9]. At the

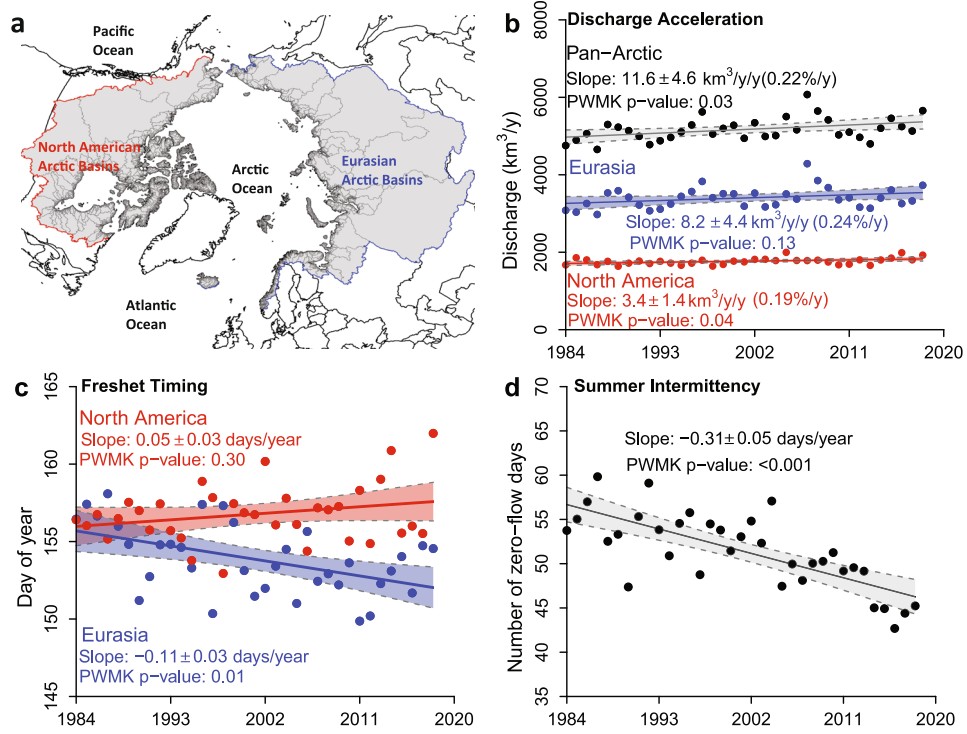

**Fig. 1 Changes in total water export, freshet timing, and summer stream intermittency. a** Map showing the pan-Arctic (gray shaded), North American (outlined in red), and Eurasian (outlined in blue) basins; **b** Water export from rivers shows increasing trends across the Pan-Arctic, Eurasia, and North America during 1984–2018; **c** The temporal centroid of spring freshet (TCSF) for North American and Eurasian rivers per year shows a significant advance of the TCSF for Eurasia. Data points represent the mean TCSF of all reaches in each region for 1984–2018; **d** The number of zero-flow days (ZFD) during the open-water period (Apr–Nov) has decreased significantly by 3.1 days/decade for streams prone to intermittency during 1984–2018. Each data point represents the annual average ZFD for all intermittent reaches across the pan-Arctic. The shade in **b**, **c**, and **d** indicates the 95% confidence interval. See Methods for the definitions of total water export, TCSF, ZFD, and intermittent reaches. PWMK refers to the pre-whitening Mann–Kendall test for trend significance.

basin scale, we found the upstream/middle regions of the Yukon and Mackenzie River basins (e.g., Canadian Shield) had a noticeable decrease in annual discharge, while the rivers draining into the Hudson Bay show a more pronounced increase than other North American Arctic basins. In Eurasia, the trends in river discharge also show substantial regional variability; for example, river discharge has substantially decreased in the upper Yenisey but increased in the central region of the Lena basin (Fig. 2). Temporal trends in discharge also show differences across ecoregions and permafrost regimes (irrespective of basin, Supplementary Fig. 3). For example, desert and semi-desert (Supplementary Fig. 3a) and continuous permafrost regions (Supplementary Fig. 3b) show more increased flows across all basins, while steppes (Supplementary Fig. 3a) show sharply decreasing flows.

Moving beyond the water balance, we also find significant changes in the spring freshet and river intermittency (Fig. 1c, d). We calculated the temporal centroid of spring freshet (TCSF), which is the temporal centroid of river discharge for March–July (See "Methods" for details), to reflect the timing of the spring freshet. North American basins showed no significant trends in TCSF, while Eurasian basins saw earlier freshet at the rate of 1.1 days/decade (Fig. 1c), which literature suggests is likely related to reservoir operations[40] and observed trends in earlier snowmelt in Eurasian basins[15,41]. We calculated the number of zero-flow days (ZFD) during April–November for each reach to quantify open-water stream intermittency, defined as the number of days with a daily mean discharge of <0.001 m³/s during

April–November. We found that the average ZFD of all inter-mittent reaches (i.e., 91,663 or 19% of total reaches, see Methods for the definition of an intermittent reach) in the pan-Arctic shows a strong decline at a rate of 3.1 days/decade (Fig. 1d). This decline means that intermittent river reaches are getting wetter and are running dry less often[42].

Since RADR takes on information from both remote sensing and hydrologic modeling, we analyzed the individual contributions of RADR's two components (i.e., modeled hydrologic reanalysis and Landsat remote sensing) to trace the provenance of our updated Arctic discharge understanding and attribute these changes in understanding. The modeled reanalysis products used in RADR have existed since 2019[25] and 2020[26], but neither has been analyzed or compared with other studies in the Arctic. We found that assimilating satellite data into the hydrologic models significantly improved the accuracy of modeled daily discharge at validation gauges (Fig. 3 and Supplementary Figs. 2 and 4) and altered understanding of Arctic rivers across the entire domain (compare Figs. 1, 2 and Supplementary Figs. 5, 6). We assessed the error metrics Nash Sutcliffe Efficiency[43] (NSE) and Kling-Gupta efficiency[44] (KGE) of both RADR and the baseline models at 1079 validation gauges at the daily scale, calculated over coincident times with in situ observations during 1984–2018 (Fig. 3). We found that median NSE and KGE increased by 0.16 and 0.09 across these gauges, respectively, when comparing RADR and the original model simulations (Supplementary Fig. 4a, b), and this median improvement is larger for regulated reaches (Supplementary Fig. 4c, d), showing the value of remote

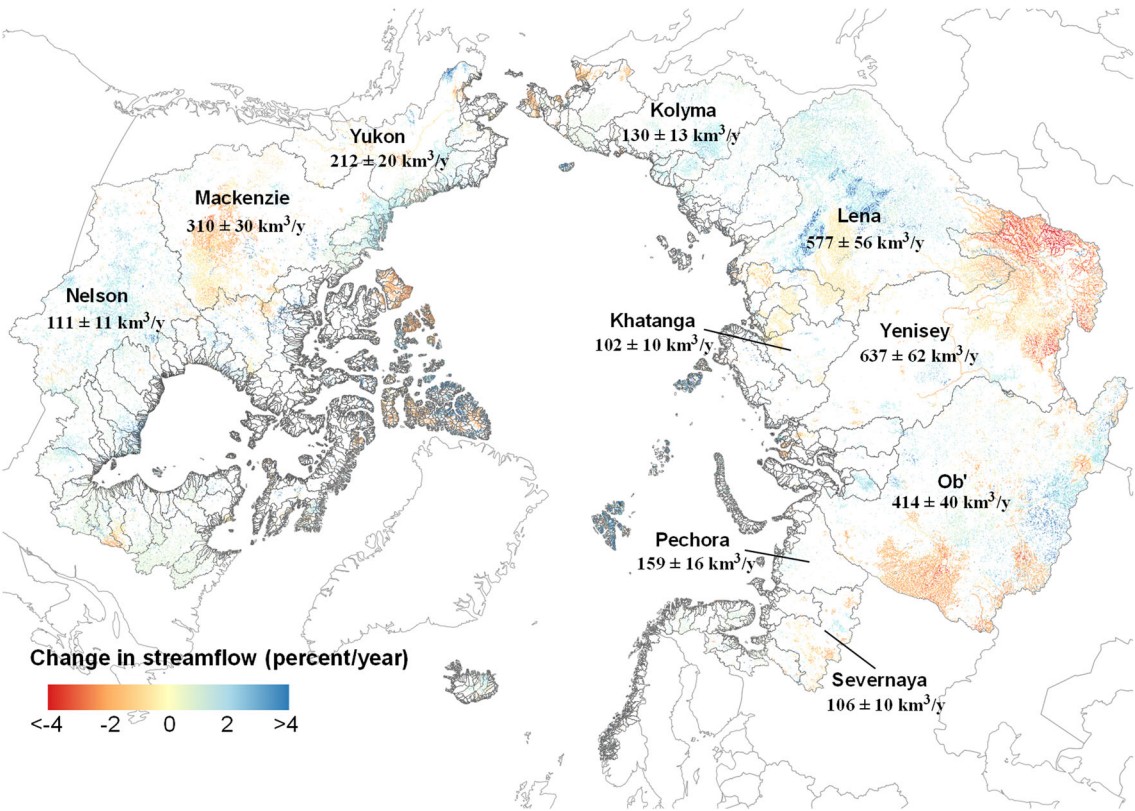

**Fig. 2 Temporal trends in river discharge during 1984–2018.** Arctic rivers show substantial spatial variations in discharge trends. Only rivers with statistically significant trends are mapped (percent/year, p-value<0.05). Trend significance is calculated using the pre-whitening Mann–Kendall test. The mean annual discharge (km³/y) is labeled for ten major river basins (see more details about its change in Supplementary Table 2).

sensing in 'seeing' impacts of human regulations. We note that RADR's water balance is derived from the models and its improvements are largely in dynamics. For example, the performance of RADR is sometimes much improved, but still with low absolute skill and sometimes large biases, especially in regulated reaches (Supplementary Table 2 and Supplementary Fig. 7). RADR was also more accurate than the baseline models across the entire domain, not just in aggregate (Fig. 3, Supplementary Fig. 2). In particular, RADR's accuracy increases with increasing stream order (i.e., size), but RADR's improvements over baseline models are greatest at smaller rivers (Supplementary Fig. 2). Recall those small rivers dominate Arctic hydrography and are the least gauged, and RADR makes the largest and most accurate changes in exactly these reaches (Supplementary Figs. 2 and 8).

Summarizing the noticeable differences at RADR's base resolution of daily flows in individual reaches, the difference in total water export to the ocean between RADR and the baseline models is negative 4% (Supplementary Fig. 8), and the median relative bias of RADR is negative 2.5% compared with in situ observations (Supplementary Figs. 2 [daily] and 9 [monthly, including an additional 1,077 monthly gauges and aggregating daily gauges to monthly]). This suggests that addition of satellite data to the models has actually decreased the discharge in the original models in aggregate, and our estimates of the bulk water balance and acceleration are likely conservative, despite RADR showing substantial and significant increases in discharge against previous understandings of Arctic hydrology in the literature[29]. We can thus attribute the updated understanding of Arctic rivers in RADR mainly to the improved modeling capabilities and their forcings[9,10]. However, Fig. 3 and Supplementary Figs. 2 and 9 show that relying on these models alone to provide pan-Arctic understanding would estimate too large of an increase in water

balance and would introduce reach-level errors in the freshet timing, river intermittency, and associated distributions across ecoregion/permafrost zones (Supplementary Figs. 5 and 6). There are other hydrologic models and different forcing combinations we could have used for our baseline product, and changing the models would change the assimilated results. However, our ensemble of two recent, global, and crucially publicly available models, each with a different forcing, provides a level of robustness commensurate with previous model efforts in the region. Ultimately, by systematically evaluating RADR against all publicly available data (Fig. 3 and Supplementary Figs. 2, 4, and 9), we show that we have more accurately estimated recent changes to Arctic rivers by invoking primary data collected from outer space rather than relying on gauges or models alone.

We have revealed a new scope of recent changes in the Arctic rivers, which has important implications. For example, our finding of further increases in the quantity of Arctic river discharge 1984–2018 suggests that previous assessments of biological productivity, Arctic societal water supply, and the freshening of the Arctic Ocean can be updated with more accurate and complete inputs. Further, RADR's spatial completeness allows us to show substantial regional and local differences, highlighting the importance of providing discharge at the reach scale. The updated understanding of recent changes to Arctic river discharge revealed here should subsequently improve our ability to project the future of the Arctic.

RADR builds on a decade of advances in remote sensing and hydrologic modeling and shows that remotely sensed data can significantly improve the accuracy of discharge reanalysis. Given that the hydrologic models and Landsat data used to produce RADR are publicly available for all rivers on earth, we suggest that analogous updates in hydrologic understanding might be

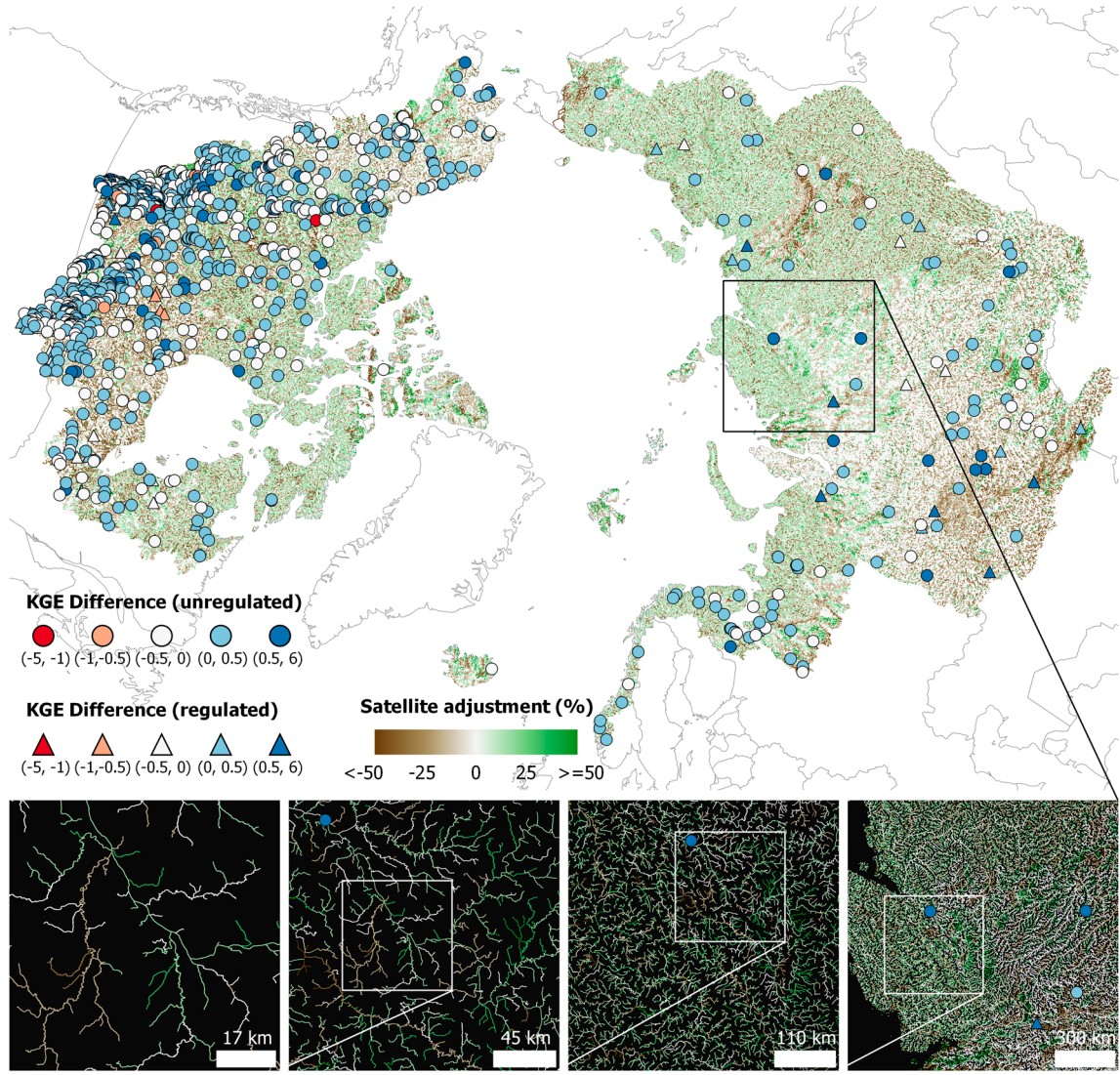

**Fig. 3 Discharge differences attributed to satellite data.** The satellite adjustment (relative difference, %) between RADR and baseline models in the annual mean discharge at each of 486,493 reaches is shown in a color gradient from brown (less water) to green (more water). Colored circles (for unregulated reaches) and triangles (for regulated reaches) indicate the difference in KGE (Kling-Gupta efficiency, a standard error metric, positive difference indicates improvement) of daily discharge between RADR and baseline model simulations evaluated against 1,079 gauges at the daily scale. See Methods for the definition of the regulated reach and baseline model simulations. The four boxes at the bottom show spatial details of the difference.

gained in the rest of the globe following our methods to potentially improve our understanding of the global hydrologic cycle. We reiterate that neither gauges, models, nor remote sensing alone can provide the richest and most accurate picture of Arctic rivers, and that only by combining these data did this updated understanding of recent changes to the Arctic river discharge emerge.

## Methods

In this study, we developed a river discharge reanalysis product for the pan-Arctic region (RADR), which provides daily discharge for 486,493 river reach segments for 1984–2018. We generated this product by assimilating approximately 9.18 million discharge observations derived from 227 million width measurements made from 155,710 Landsat images into the hydrologic model simulations.

**Data**. We used the following data to perform this study.

1. Multi-Error-Removed-Improved-Terrain (MERIT) DEM[22]
2. MERIT Hydro[21]
3. MERIT-Basin hydrography[25]
4. Global River Widths from Landsat (GRWL)[20]
5. HydroLAKES[45]

6. Discharge data from the U.S. Geological Survey (USGS, https://waterdata.usgs.gov/nwis/sw), ECCC (Environment and Climate Change Canada, https://www.canada.ca/en/environment-climate-change.html), Centre d'Expertise Hydrique du Québec (https://www.cehq.gouv.qc.ca/), Global Runoff Dataset Center (GRDC Database), R-ArcticNET (https://www.r-arcticnet.sr.unh.edu/v4.0/index.html), and Arctic Great River Observatory (https://arcticgreatrivers.org/)
7. Landsat surface reflectance tier 1 image collections from the ETM (Enhanced Thematic Mapper), ETM + (Enhanced Thematic Mapper plus), and OLI/TIRS (Operational Land Imager/Thermal Infrared Sensor) sensors
8. The Terrestrial Ecoregions of the World (TEOW)[46]
9. The permaice data from the National Snow and Ice Data Center (https://nsidc.org/data/ggd318)[47]
10. Global Reach-level A priori Discharge Estimates (GRADES) runoff and discharge[25]
11. Global Flood Awareness System (GloFAS) discharge reanalysis[26]
12. DOR (degree of regulation)[48]
13. GlObal geOreferenced Database of Dams (GOODD) dataset[49]

MERIT DEM[22] is a global terrain elevation dataset at a resolution of 3 arcseconds. It has removed multiple errors, including absolute bias, stripe noise, speckle noise, and tree height bias, and represents the current best globally consistent topography data. We used MERIT DEM to calculate plane slopes required by the hydrologic routing model.

MERIT Hydro[21] is a global hydrography dataset developed based on the MERIT DEM. It provides high resolution (3 arcseconds) data of flow direction, elevations, drainage area, and river channel width. We downloaded the river channel width data from http://hydro.iis.u-tokyo.ac.jp/~yamadai/MERIT_Hydro/, extracted the river width at locations of river centerlines, and then calculated the mean width for each river reach segment later used to categorize rivers sizes.

MERIT-Basin hydrography[25] (Hydro_v07_Basins_v01) is a global river network vector dataset developed based on the MERIT DEM. It delineates sub-catchments and river channels with a median drainage area of 37 km$^2$ with the Pfafstetter code system[50]. It represents the highest resolution representation of the global river network currently available. It provides two vector shapefiles: sub-catchments and river centerlines. We used the centerline dataset to identify river channel locations. We also used properties of both vectors, including upstream/ downstream links, upstream drainage area, channel slope, channel length, sub-catchment area, to prepare inputs for the routing model.

GRWL[20] is a global river width database derived from Landsat images, providing 58 million river width measurements of rivers >30 m wide at annual mean discharge. We downloaded the GRWL vector data v01.01 from https:// zenodo.org/record/1297434#.XxIynyhKiUk, and spatially joined it to the MERIT-Basin river network. Then we used the width_m property in GRWL to calculate the mean width for each river reach in our study region. This mean width is further used to determine river sizes and validate our width mapping.

HydroLAKES[45] is a global database providing the shoreline polygons of 1.4 million lakes with a surface area greater than 10 ha. We downloaded the dataset from https://www.hydrosheds.org/pages/hydrolakes. We spatially joined the HydroLAKES polygons to the MERIT-Basin river network to identify reaches connecting with or as a part of lakes. This information enabled us to identify river reaches suitable for remote sensing of discharge.

The daily discharge data (i.e., river flowrate, unit: ft$^3$/s or m$^3$/s) collected from the U.S. Geological Survey (USGS, https://waterdata.usgs.gov/nwis/sw), ECCC (Environment and Climate Change Canada, https://www.canada.ca/en/ environment-climate-change.html), Centre d'Expertise Hydrique du Québec (https://www.cehq.gouv.qc.ca/), Global Runoff Dataset Center (GRDC Database), R-ArcticNET (https://www.r-arcticnet.sr.unh.edu/v4.0/index.html), and Arctic Great River Observatory (https://arcticgreatrivers.org/) are used to calibrate and validate our model and discharge product. Monthly data from the R-ArcticNET are also collected for monthly evaluation of our product. Specifically, we converted the spatial locations (longitude and latitude) of these discharge gauges into point shapefiles and then matched them with the MERIT-basin hydrography based on their locations and drainage area (within 500 m from the river channel centerline and within ± 10% of drainage area between the gauge data and MERIT-Basin), resulting in 1079 effective daily gauges and 1076 monthly gauges.

The Landsat surface reflectance tier 1 image collections are the atmospherically corrected surface reflectance from the Landsat 5 (ETM), 7 (ETM + ), and 8 (OLI/ TIRS) sensors. They provide visible and near-infrared (VNIR), short-wave infrared (SWIR), and thermal infrared (TIR) bands at a 30 m resolution. The ETM/ ETM + images have four VNIR bands, two SWIR bands, and one TIR band, whereas OLI/TIRS images provide five VNIR bands, two SWIR bands, and two TIR bands. We accessed these images for 1984–2018 from the collections of LANDSAT/LT05/C01/T1_SR, LANDSAT/LE07/C01/T1_SR, and LANDSAT/ LC08/C01/T1_SR archived in Google Earth Engine (GEE)[51]. We used the VNIR and SWIR bands to classify water pixels, which are further used for river width calculation.

TEOW[46] is a global map that regionalizes earth's terrestrial biodiversity into 867 ecoregions, which are further classified into 14 distinct biomes, such as forests, taiga, and tundra. We intersected this vector dataset with the sub-catchment polygons of the MERIT-basin hydrography to determine the dominant biome of each sub-catchment.

The permaice data from the National Snow and Ice Data Center[47] describes the properties of permafrost in the Arctic region. It groups the permafrost into four major categories: continuous, discontinuous, sporadic and isolated, with four subcategories in each based on the ground ice content and thickness of overburden. We downloaded the permaice data from (https://nsidc.org/data/ggd318). Similar to TEOW, we intersected the permafrost polygons with the MERIT Basin sub-catchments, which was used to determine the dominant permafrost for each sub-catchment.

GRADES[25] runoff is a recently published global surface runoff product with a resolution of 0.25° for 1979–2013. It was generated by the Variable Infiltration Capacity (VIC) model forced with the global precipitation dataset MSWEP V2 (Multi-Source Weighted-Ensemble Precipitation, version 2) with resolutions of 0.1° and 3 h[52] and climate variables from the Climate Forecasting System Reanalysis[53] and calibrated/bias-corrected with a unique set of global runoff signature maps[25]. We extended the original GRADES simulations for recent years (2014–2018) using the same approach with MSWEP V2 precipitation dataset and climate variables from ERA5 (https://www.ecmwf.int/en/forecasts/datasets/reanalysis-datasets/era5). Hereafter, we refer to extended GRADES simulations performed here and original GRADES together as 'GRADES.' We have conducted a systematic evaluation against a consistent forcing and confirmed that this approach can consistently represent the temporal variability during the entire study period.

GloFAS discharge reanalysis[26] is another recently developed global river discharge product, which provides daily river discharge for global rivers (mapped

at 0.1° grid resolution) for 1980–2018. It is publicly available at https:// data.jrc.ec.europa.eu/collection/id-00288. Similar to GRADES, we use GloFAS discharge product as a baseline model into which we assimilate the remotely sensed discharge. Since both GRADES and GloFAS are recently available global river discharge reanalysis products, we took them as the model baselines and applied our data assimilation scheme to both of them to develop an optimal discharge product. We make an optimal blend of these two products to produce a single 'baseline', which will be described later.

DOR[48] is a recently published reach-based product that provides the degree of regulation for global river reaches. We use this dataset to define regulated reaches in our study.

GOODD[49] is the latest georeferenced data of global dams. We use this dataset to identify river reaches impacted by dams.

**Width extraction**

*Filtering river reaches suitable for width extraction.* The Landsat program offers a succession of satellites with varying capabilities, but 30 m multispectral images suitable for water detection are common to each satellite dating back to 1984. Landsat imagery cannot reliably measure changing widths on reaches narrower than 90 m with a resolution of 30 m[20]. Therefore, before extracting river widths from these images, we filtered MERIT Basin river reaches by removing those narrower than 90 m. It is important to note that we simulate discharge in rivers narrower than 90 m via data assimilation, described later, but 90 m was our remote sensing limit. To identify these reaches, we first extracted the mean river width for each river reach segment by spatially joining the GRWL and MERIT Hydro width datasets with MERIT Basin river network, and then calculated the mean GRWL width ($W_{GRWL}$) and mean MERIT Hydro width ($W_{MERIT}$) for each reach where available. We selected reaches satisfying the following criteria: max($W_{MERIT}$, $W_{GRWL}$) > 90 m. Here, we used both GRWL and MERIT Hydro to maximize the spatial coverage, as we found neither of them completely covers the MERIT Basin river reaches. We also removed reaches that directly connect with or are included as a part of lakes, which were identified with the HydroLAKES data, as the discharge inversion algorithm used in this study (i.e., Bayesian AMHG-Manning's algorithm, geoBAM) is not suitable for such reaches.

*Constructing orthogonal vectors.* After determining the river centerlines suitable for width extraction, we need to select cross-section locations in the reach to extract river width, since geoBAM requires multi-temporal widths at multiple cross-sections along a reach. Here, we used a dynamic procedure to locate width measurements due to computational limits at the global scale and due to practical expediency (i.e., there is no need to measure widths every 30 m as in previous work to estimate discharge on a 2 km wide river). Previous studies[24,54] have shown sufficient distance between cross-sections is needed to capture the geomorphic variability, a requisite of geoBAM, especially for large rivers. We arranged the cross-sections for each reach dynamically based on their average widths (i.e., mean($W_{MERIT}$, $W_{GRWL}$)) (Supplementary Table 3). This alignment ensures a balance between computational efficiency and data density while maintaining the requirement of geoBAM. To reduce errors in extracted widths, we removed cross-sections close to river confluences, as these cause known errors in the width extraction process[24].

We used the RivWidthCloud method[55] to extract multi-temporal river widths from the Landsat family of satellites for the period 1984–2018 in GEE. To estimate river width, RivWidthCloud maps a fixed orthogonal vector onto a river centerline and then extracts the water area under each orthogonal per image to report river width. If orthogonals are too long, classified water out-of-bank might introduce a false positive, while if orthogonals are too short, they will not span the full river width. To determine orthogonal length, we calculated the temporally variable distance between the center and boundary of the river flow extent (bank distance, or BD) for each cross-section using the CalcDistanceMap function in the RivWidthCloud method. Then, we used the following formula to determine the length of the orthogonal line for each cross-section.

$$L_{orth} = \max(BD_{75} \times 2, W_m) \times 2 \tag{1}$$

Where $L_{orth}$ is the length of the orthogonal line, $BD_{75}$ is the 75 percentile bank distance; $W_m$ is max($W_{MERIT}$, $W_{GRWL}$).

We then used the CalculateAngle function in the RivWidthCloud method to estimate the angle of the orthogonal lines.

*Extracting river width.* With orthogonals defined, we used the revised Dynamic Surface Water Extent (DSWE) method developed by the USGS[56] to classify water pixels of Landsat images with cloud coverage of less than 25% in GEE. Water classification from Landsat has a long history and numerous methods, but DSWE is a recent innovation that is accurate and computationally efficient. We applied this method to images during April–November to ensure the accuracy of classified water mask and resulting width estimates.

Our process resulted in 227 million width observations for 2.93 million cross-sections in 131,153 reaches, made from 155,710 Landsat images. This large volume of earth observation data forms the basis of discharge estimation and serves as 'evidence' for improving hydrological knowledge. It is precisely this volume of

extant satellite data we contend exist for many places on earth that can be leveraged for improving our knowledge of past hydrologic states.

We evaluated our width extraction process by comparing the mean of the widths obtained in this study with GRWL and found the relative bias is only 0.6%, suggesting a satisfactory accuracy of our method. GRWL was extensively validated where possible, as ground-truth data at this scale simply do not exist. We believe the comparison to GRWL is an acceptable indication of width accuracy, but we acknowledge that we cannot validate our multi-temporal widths at this scale.

**Simulating daily discharge**

*Remote sensing of discharge.* We used geoBAM[19] to invert river discharge from river width time series extracted from satellite imagery. geoBAM estimates river discharge in a probabilistic manner on the basis of at-many-stations hydrologic geometry (AMHG)[17].

$$\log W_i = b_i(\log Q - \log Q_c) + \log W_c + \epsilon \qquad (2)$$

where $W_i$ is river width at cross-section $i$, (m); $b_i$ is the AHG width-discharge exponent at cross-section $i$. $Q$ is river discharge (the desirable quantity), (m³/s); $Q_c$ and $W_c$ are AMHG global parameters for a reach; $\epsilon$ is an error term. By implementing an efficient parallel-chain sampler, geoBAM estimates $Q$, $Q_c$, and $W_c$ through Bayesian inference.

AMHG characterizes a semi-log linear relationship between the coefficients and exponents in traditional at-a-station hydrologic geometry (AHG, $w = aQ^b$, where $w$ is width, and $Q$ is discharge) at multiple cross-sections along a river reach. Since AMHG links width to discharge over both time and geographic space, it reduces the number of unknown parameters of traditional AHG, thus showing an appealing advantage for remote sensing discharge estimation, especially for ungauged basins.

One advantage of geoBAM is that it does not require ground-based in situ measurements of any kind to estimate discharge, although such data are beneficial to improve accuracy as geoBAM relies on discharge priors as a Bayesian technique. These priors can come from field measurements or modeled discharge. Another vital aspect of geoBAM is that it also provides the uncertainty associated with each discharge estimate, paving the way for further assimilating such estimates into hydrologic model simulations. Ref. [19] developed geoBAM by incorporating refined prior information constrained by river geomorphology through a machine learning approach. With more constrained a priori data, geoBAM substantially improves the accuracy of estimated discharge[19]. Consult refs. [18,19] for more information about geoBAM.

geoBAM runs on a maximum of 40 cross-sections per reach for computational efficiency. When reaches had more than 40 cross-sections (there are 21,055 out of 131,153 reaches having more than 40 cross-sections), we ranked them by the number of their width observations in time and selected the top 40. In addition to multi-temporal widths, geoBAM also requires prior information, including channel slope, and min, max, and mean discharge at the reach scale. We took the channel slope from the properties of MERIT Basin hydrography. We calculated the mean, maximum, and minimum discharge based on hydrologic model simulations (see "Hydrologic model"). To minimize potential errors induced by geomorphic changes, we applied geoBAM to multi-cross-sectional width time series every five years. This means the discharge prior was calculated every five years. To improve the accuracy of geoBAM estimates, we calculated the prior over the same months when Landsat widths are available. For example, if the Landsat widths are only available for April–November, which is true for most Arctic rivers as we limited our satellite data to open-water periods, we used hydrologic model simulations of April–November to calculate the discharge prior. For the reaches where in situ discharge measurements are available, we used these data to estimate this prior information. Thus, unlike previous geoBAM studies, we here use available in situ data and hydrologic model simulations to provide priors to geoBAM in pursuit of the most accurate Arctic discharges possible. In this process, we made 9.18 million discrete discharge estimates across 131,153 reaches for the period 1984–2018.

geoBAM discharges themselves, while successfully obtained, are not sufficient to provide a comprehensive reanalysis of discharge at pan-Arctic scales: we only have geoBAM estimates at 131,153 out of 486,493 reaches and at 72 out of 12,784 discrete days, on average, per reach. Thus, we need to couple geoBAM to a hydrologic model to provide needed daily flow estimates for the pan-Arctic. Additionally, while the accuracy of geoBAM is acceptable in aggregate, there are reaches and times for which geoBAM discharge estimates are inaccurate. Thus, the uncertainty in geoBAM estimates should be considered when combining geoBAM with a hydrologic model. This need is tailor-made for data assimilation.

*Hydrologic model.* To simulate daily discharge in the pan-Arctic, we used GRADES and GloFAS products to generate baseline simulations. Both GRADES and GloFAS are recently developed global discharge products. Using both of them as baseline models enabled us to leverage the strength of these two products and develop an improved discharge reanalysis product. For GRADES, we extracted the area-weighted average runoff for each sub-catchment, which was then laterally routed to the channels and integrated along the river network using the Hillslope River Routing (HRR) model[57,58]. GloFAS is a gridded product that needed to be transformed into our explicit vector river network from the MERIT Basin hydrography. For GloFAS, we first estimated the gridded runoff by dividing the

discharge by the drainage area. Then we extracted GloFAS runoff for each sub-catchment in the MERIT Basin hydrography based on spatial locations and drainage area. Then we routed the GloFAS runoff with HRR through the MERIT Basin river network. In this process, we generated two sets of hydrologic model simulations (GRADES-based and GloFAS-based) for the same hydrography. Then we assimilated our remotely sensed river discharge into each of them and chose the final reanalysis results based on their performance (see "Data assimilation" for details).

HRR routing is central to our assimilation and modeling. It requires the upstream drainage area, sub-catchment area, and channel and plane length/slope for each sub-catchment as inputs. We obtained all these parameters save the plane length and slope from the properties of MERIT Basin hydrography. We calculated plane length by dividing the sub-catchment area by the channel length based on the 'open-book' assumption in HRR[57]. For the plane slope, we first calculated the terrain slopes using the tauDEM software[59] with the MERIT DEM[22] as input, and then took the mean as the plane slope for that sub-catchment. The channel routing simulation in HRR also requires the bankfull width and reference flow rate for each model unit. We calculated bankfull width as the 90th percentile river width extracted from Landsat images representative of river width extent of 2 year flood events, which is often used to determine the bankfull width[60]. For small rivers where Landsat widths, GRWL width, and MERIT Hydro width are not available, we estimated the bankfull width based on the width-drainage area relationship from ref. [61]. Then we calculate the reference flow rate using the width-discharge equation developed by ref. [61]. The plane roughness coefficient, a parameter required for plane routing in HRR, is unknown. We estimated this parameter through a calibration approach. We calibrated this parameter against daily discharge observations from 1079 gauges for 1984–1998 and validated it for 1999–2018. Please note that more gauges have temporally complete data during the calibration period than the validation period. For example, 469 gauges are more than 90% temporally complete during 1984–1998; in contrast, this number is 208 for 1999–2018. This further confirms the decline of gauge data in recent years and highlights the necessity of our study. Routing GRADES and GloFAS surface runoff through the river network provided two sets of daily discharge products for the study region, which will be further used for data assimilation. In this study, we did not simulate the effects of reservoir operations, and lakes and reservoirs were treated as 'flat river reaches' with very low slope[62].

*Data assimilation.* To assimilate geoBAM discharge into the hydrologic model, we adopted the Local Ensemble Transform Kalman Filter (LETKF)[63]. LETKF is a widely used data assimilation algorithm in hydrology since it can deal with highly non-linear model physics inherent to most hydrological processes. The LETKF optimizes model estimation by accounting for the current status and the uncertainties of both the model predictions and the observations (here observations are geoBAM discharge estimates) (Eqs. (3)–(4)). The uncertainty of the model predictions was estimated using an ensemble approach. We generated 20 model predictions using a Monte Carlo approach, by which we randomly generated 20 coefficients in the range of 0.1–2.5 and then multiplied the baseline simulation by these coefficients to obtain an ensemble of 20 samples. We then took the ensemble deviations as prediction uncertainty. Uncertainty of geoBAM estimates was obtained from its sampling ensemble statistics as a Bayesian technique with explicitly defined uncertainty. The LETKF applies a sequential filter to local regions where observations are available to improve computational efficiency. We defined the local regions as hydrologically connected channels with a total drainage area less than a threshold (i.e., the local patch size, 5000 km² in this study).

$$\bar{\mathbf{x}}_n^a = \bar{\mathbf{x}}_n^b + \mathbf{P}_n^a \mathbf{H}_n^T \mathbf{R}_n^{-1}\left(\mathbf{y}_n^o - \mathbf{H}_n \bar{\mathbf{x}}_n^b\right) \qquad (3)$$

$$\mathbf{P}_n^a = (\mathbf{I} + \mathbf{P}_n^b \mathbf{H}_n^T \mathbf{R}_n^{-1} \mathbf{H}_n)^{-1} \mathbf{P}_n^b \qquad (4)$$

where $\bar{\mathbf{x}}_n^a$ is the analysis state estimate (i.e., discharge) at time $t_n$; $\mathbf{P}_n^a$ is the covariance of $\bar{\mathbf{x}}_n^a$; $\bar{\mathbf{x}}_n^b$ is the background state estimate (i.e., simulated discharge by the model); $\mathbf{y}_n^o$ is the observation (i.e., remotely sensed discharge); $\mathbf{H}_n$ represents the relationship between the system state $\mathbf{x}_n$ and the observations $\mathbf{y}_n^o$ at time $t_n$, and is the identity matrix in this study because both the system state and the observations are discharge here; $\mathbf{R}_n^{-1}$ is the variance of the observations.

To augment the effects of remotely sensed discharge, we applied a centered smoother at a 7 day window, assuming the ensemble analysis perturbation weights calculated at time $t$ are valid for the period $[t - 3, t + 3]$ based on the river flow wave travel time[62]. When there are other observations during the smoother window, we truncated the smoother window at the center between $t$ and the most recent observation on each side. For example, at time $t$ we have an observation (geoBAM estimate) for reach $i$, then we obtained the perturbation weights by implementing the LETKF; if there are no other observations during $[t - 3, t + 3]$, we apply the weights to the whole window period. If there are observations at, for example, $t - 3$ and $t + 2$, respectively, we will apply the weights obtained at $t$ to the window $[t - 2, t + 1]$. This approach ensures that the most recent observation is used to inform the model prediction. The flowchart of the data assimilation process can be found in Supplementary Fig. 10.

We obtained two sets of reanalysis discharge products by applying this data assimilation scheme to each of the two hydrologic model baselines (see "Hydrologic model" for details). Then we evaluated these two products by

comparing them against gauge observations. To create an optimized final result, we selected the assimilated model with better performance at the gauged reach and propagated this decision for all upstream river reaches until the next gauge is available. For river reaches in ungauged basins, we selected the one with better performance in the closest gauged basin. In this process, we generated the final reanalysis discharge: Remotely-sensed Arctic Discharge Reanalysis (RADR). To evaluate RADR, we both validated it at available gauges and compared it with the baseline model simulations. Since RADR is an optimal blend of two data assimilation results (i.e., GRADES-based and GloFAS-based), here, we refer to the 'baseline model simulation' as a blend of GRADES and GloFAS, which has the same composition as RADR. For example, if the RADR data assimilation result for a reach is based on GRADES, then the baseline model simulation for this reach is the GRADES discharge. The difference between RADR and baseline discharge should reflect the performance of the method used in this study. KGE and NSE were calculated based on the daily discharge of RADR and baseline model simulations compared with observations at these gauges. This suggests that our method can significantly improve the accuracy of simulated discharge.

### Definition of terms specific to this study

*Total water export.* It is the sum of annual average discharge from the outlets of all Arctic watersheds. The watershed outlets are defined as MERIT Basin river reaches with the 'NextDownID' (a field in the attribute table) equal to zero.

*Temporal centroid of spring freshet (TCSF).* It is defined as the temporal centroid of river discharge for the springtime March–July. In Fig. 1c, we calculated the TCSF for each reach each year, and then took an average of TCSF of all reaches in the region of interest (i.e., North America or Eurasia) for each year.

$$ \text{TCSF} = \frac{\sum_{t=1,March}^{t=31,July} \text{doy}_t \times Q_t}{\sum_{t=1,March}^{t=31,July} Q_t} \tag{5} $$

Where $\text{doy}_t$ is the day of the year at time $t$; $Q_t$ is mean daily discharge at time $t$, m$^3$/s.

*Intermittent reach.* We first calculate the number of zero-flow days (ZFD) during the open-water period (April–November) for each reach each year, and then calculate the annual average ZFD (ZFD$_{open-water}$) for each reach over the study period. We define open-water intermittent reaches as those with 7< ZFD$_{open-water}$ < 230. The zero-flow days are defined as times when the mean daily discharge is <0.001 m$^3$/s. In Fig. 1d, we calculated the ZFD for each reach each year, and then identified the open-water intermittent reaches based on the definition above, and took an average of ZFD of all intermittent reaches in the whole study region for each year.

*Regulated reach.* We use DOR[48] and GOODD[49] to quantify human regulations on rivers. We define 'regulated reaches' as those with 'DOR' > 0 or close to a GOODD dam (distance < 500 m).

### Data availability

GRADES products are previously published and publicly available at http://hydrology.princeton.edu/data/mpan/GRADES/. GloFAS products are previously published and publicly available at https://data.jrc.ec.europa.eu/collection/id-00288. RADR is available at Zenodo: https://doi.org/10.5281/zenodo.5604980.

### Code availability

The RivWidthCloud[23] code for Landsat river width extraction is available at https://github.com/seanyx/RivWidthCloudPaper; The geoBAM[19] algorithm used in this study is available at https://github.com/craigbrinkerhoff/geoBAMr; The data assimilation[27] package is available at https://github.com/Fluvial-UMass/SIRD_Missouri.

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

## Acknowledgements
We thank Tamlin M. Pavelsky at UNC Chapel Hill for constructive feedback on a draft version of this manuscript, and acknowledge and thank ECMWF for making GloFAS data publicly available, Edward Beighley for developing HRR, and Craig Brinkerhoff for developing geoBAM. D. F. was supported by NASA New Investigator Grant 80NSSC18K0741 awarded to C.J.G. and C.J.G. was partially supported by NASA SWOT Science Team grant 80NSSC20K1141 and NSF CAREER grant 1748653.

## Author contributions
D.F. conducted all analysis and wrote the original manuscript; C.J.G. conceived the study, supervised the project, and revised the manuscript; P.L. provided GRADES products and assisted with the routing model setup and GRDC data collection; X.Y. assisted with RivWidthCloud modification and data visualization; M.P. provided VIC runoff and discharge products; Y.I. developed the data assimilation algorithm. All authors edited the manuscript and participated in revisions.

## Competing interests
The authors declare no competing interests.
