## [Peer Review File · Nature Communications]

Recent changes to the Arctic river dischargeEditorial Note: This manuscript has been previously reviewed at another journal that is not operating a transparent peer review scheme. This document only contains reviewer comments and rebuttal letters for versions considered at Nature Communications.

Reviewers' Comments:

Reviewer #1:

Remarks to the Author:

Re-Review of "Recent changes to Arctic river discharge" by Dongmei Feng, Colin J. Gleason, Peirong Lin, Xiao Yang, Ming Pan, and Yuta Ishitsuka

Submitted to Nature Communications

Manuscript Number: NCOMMS-21-34192-T

Summary: In this paper, the authors explore the 1984-2018 pan-Arctic river discharge using a novel method titled "Remotely-sensed Arctic Discharge Reanalysis" or RADR. This method assimilates remote sensing imagery (Landsat optical imagery at 30-m resolution) into two hydrological models to obtain daily streamflow time series at 486,493 Arctic river reaches. RADR yields pan-Arctic discharge estimates of 5,169 km³ yr⁻¹ over 1984-2018, in line with prior studies (e.g. Shiklomanov et al. 2021). Of note is the recent acceleration of inflows to the Arctic Ocean basin, with increases of 11.6 km³ yr⁻¹ over the 35-year study period.

I sincerely thank the authors for addressing in a satisfactory manner my earlier comments on a previous version of their manuscript that was submitted to Nature. Their detailed responses to those comments and the additional results presented in the paper suggest that it should now be considered for publication in Nature Communications. While observational data for river discharge in Eurasia remain sparse relative to data availability in North America, they instill greater confidence in the results over this vast portion of the pan-Arctic domain.

General Comments:

- 1) Given the paper has been transferred from Nature to Nature Communications, it will require some modifications to follow the formatting guidelines for the journal. For instance, the abstract should not exceed 150 words and exclude references, the paper should have sections and subsections where appropriate, and each requires specific content and has word count limitations.
- 2) Augmentation of the observational database of river discharge in Eurasia certainly fills a major prior gap that previously existed in this study. Thus, many thanks to the authors for the inclusion of 69 additional gauges for Eurasia sourced from R-ArcticNet; nevertheless, there is still a disproportionate amount of gauges in North America relative to Eurasia. Perhaps one recommendation for future work that could be put forward is to further evaluate the proposed methodology across the Eurasian domain to ensure it is entirely robust across all of the pan-Arctic.

Specific Comments:

- 1) P. 2, line 83: Note spelling mistake in "assimilation".
- 2) P. 3, line 89: Insert a comma in "6,073".
- 3) P. 6, line 168: It should be up to the readers, not the authors, to make the judgement whether the results presented in the paper truly are "leaps of knowledge".
- 4) P. 10, Extended Data Figure 1: Insert "(°N)" after "Latitude" on the x-axis labels in panels A and B. There is a bias in the number of Landsat images from 2013 onward relative to prior years, would this influence the results in any way?
- 5) P. 12, Extended Data Figure 3: It is unclear how changes in streamflow are classified according to ecoregions and permafrost distribution. For instance, tundra is assigned a value of 25.3%, but a gauge lying over Arctic tundra may include a catchment that also drains the boreal forest. Some clarity is needed as to how these percentages are computed and what they actually represent.
- 6) P. 23, Extended Data Table 2: For clarity, perhaps specify the data reported in this table are based on RADR. Please also insert "Arctic" before "Archipelago" in the caption.

- 7) P. 29, line 573: Ensure variables "w" and "Q" are in italics.
- 8) P. 30, line 608: Delete "in order".
- 9) P. 34, line 758: The journal is missing for reference # 14.
- 10) P. 34, line 767: Use upper case first letter in "Science".
- 11) P. 35, line 784: Perhaps other details like the publisher are needed here.
- 12) P. 36, line 817: Add details on this publication like the volume number and page range.
- 13) P. 36, line 822: Add the article # or page range for this reference.

We have addressed the referee's comments in the revised manuscript. We have provided a point-by-point response to the comments in this document.

REVIEWERS' COMMENTS

Reviewer #1 (Remarks to the Author):

Re-Review of "Recent changes to Arctic river discharge" by Dongmei Feng, Colin J. Gleason, Peirong Lin, Xiao Yang, Ming Pan, and Yuta Ishitsuka

Submitted to Nature Communications

Manuscript Number: NCOMMS-21-34192-T

Summary: In this paper, the authors explore the 1984-2018 pan-Arctic river discharge using a novel method titled "Remotely-sensed Arctic Discharge Reanalysis" or RADR. This method assimilates remote sensing imagery (Landsat optical imagery at 30-m resolution) into two hydrological models to obtain daily streamflow time series at 486,493 Arctic river reaches. RADR yields pan-Arctic discharge estimates of 5,169 km³ yr⁻¹ over 1984-2018, in line with prior studies (e.g. Shiklomanov et al. 2021). Of note is the recent acceleration of inflows to the Arctic Ocean basin, with increases of 11.6 km³ yr⁻¹ yr⁻¹ over the 35-year study period.

I sincerely thank the authors for addressing in a satisfactory manner my earlier comments on a previous version of their manuscript that was submitted to Nature. Their detailed responses to those comments and the additional results presented in the paper suggest that it should now be considered for publication in Nature Communications. While observational data for river discharge in Eurasia remain sparse relative to data availability in North America, they instill greater confidence in the results over this vast portion of the pan-Arctic domain.

Response: *Thanks to the reviewer for these comments, and we feel the manuscript has improved as our dialogue has evolved.*

General Comments:

1) Given the paper has been transferred from Nature to Nature Communications, it will require some modifications to follow the formatting guidelines for the journal. For instance, the abstract should not exceed 150 words and exclude references, the paper should have sections and subsections where appropriate, and each requires specific content and has word count limitations.

Response: *We have modified the format throughout the manuscript according to the formatting guidelines.*

2) Augmentation of the observational database of river discharge in Eurasia certainly fills a

major prior gap that previously existed in this study. Thus, many thanks to the authors for the inclusion of 69 additional gauges for Eurasia sourced from R-ArcticNet; nevertheless, there is still a disproportionate amount of gauges in North America relative to Eurasia. Perhaps one recommendation for future work that could be put forward is to further evaluate the proposed methodology across the Eurasian domain to ensure it is entirely robust across all of the pan-Arctic.

Response: *Thanks for this recommendation. We will incorporate this idea into our future work, and it would be great if this publication pushes the community toward more public sharing of daily data in Eurasia as a means to improve models and RS solutions alike.*

Specific Comments:

1) P. 2, line 83: Note spelling mistake in “assimilation”.

Response: *Corrected.*

2) P. 3, line 89: Insert a comma in “6,073”.

Response: *Corrected.*

3) P. 6, line 168: It should be up to the readers, not the authors, to make the judgement whether the results presented in the paper truly are “leaps of knowledge”.

Response: *Agreed. We have changed it to “changes in understanding”*

4) P. 10, Extended Data Figure 1: Insert “(°N)” after “Latitude” on the x-axis labels in panels A and B. There is a bias in the number of Landsat images from 2013 onward relative to prior years, would this influence the results in any way?

Response: *The figure has been modified based on the reviewer’s suggestion.*

Given the assimilation scheme, less data means that the blend of RS and model data is tipped more toward the models. Therefore, with fewer satellite data during the earlier years, it is likely that the reanalysis result is closer to the original model simulation, because less remote sensing adjustment is made. That said, with a 7 day smoother there is still ample data to change the model results even pre 2013.

5) P. 12, Extended Data Figure 3: It is unclear how changes in streamflow are classified according to ecoregions and permafrost distribution. For instance, tundra is assigned a value of 25.3%, but a gauge lying over Arctic tundra may include a catchment that also drains the boreal forest. Some clarity is needed as to how these percentages are computed and what they actually represent.

Response: *We see this point. We calculated these percentages as total percents of land cover, and each of our ~490,000 reaches therefore sits within a region that may include (especially for*

downstream reaches) a land area draining multiple types. In theory, this should be ok: the headwaters will exist in small basins with less mixed types, and as the reaches progress downstream the type will shift northward to reflect total watersheds that are mixed but reaches that are correctly classified.. The following text is added for clarification:

“...calculated as the total area of each classification divided by the total land surface area of the pan-Arctic region. Streamflow changes are attributed to the land classification at each reach, which could drain an area of mixed ecoregion/permafrost type ”

6) P. 23, Extended Data Table 2: For clarity, perhaps specify the data reported in this table are based on RADR. Please also insert “Arctic” before “Archipelago” in the caption.

Response: *It has been modified.*

7) P. 29, line 573: Ensure variables “w” and “Q” are in italics.

Response: *Done.*

8) P. 30, line 608: Delete “in order”.

Response: *Done.*

9) P. 34, line 758: The journal is missing for reference # 14.

Response: *It's a book chapter. The reference has been updated.*

10) P. 34, line 767: Use upper case first letter in “Science”.

Response: *Corrected.*

11) P. 35, line 784: Perhaps other details like the publisher are needed here.

Response: *The reference is consistent with the suggested citation information.*

12) P. 36, line 817: Add details on this publication like the volume number and page range.

Response: *Done.*

13) P. 36, line 822: Add the article # or page range for this reference.

Response: *Done.*